# Robust Vaccine-Induced as Well as Hybrid B- and T-Cell Immunity across SARS-CoV-2 Vaccine Platforms in People with HIV

Myrthe L. Verburgh,[a,b,c,d] Lisa van Pul,[b,e] Marloes Grobben,[b,f] Anders Boyd,[g,h] Ferdinand W. N. M. Wit,[a,b,g] Ad C. van Nuenen,[b,e] Karel A. van Dort,[b,e] Khadija Tejjani,[b,f] Jacqueline van Rijswijk,[b,f] Margreet Bakker,[b,f] Lia van der Hoek,[b,f] Maarten F. Schim van der Loeff,[a,b,h] Marc van der Valk,[a,b,g] Marit J. van Gils,[b,f] Neeltje A. Kootstra,[b,e] Peter Reiss,[a,b,d,i] for the AGE<sub>h</sub>IV Cohort Study

[a]Amsterdam UMC, University of Amsterdam, Infectious Diseases, Amsterdam, The Netherlands
[b]Amsterdam Institute for Infection and Immunity, Amsterdam, The Netherlands
[c]Amsterdam Public Health, Global Health, Amsterdam, The Netherlands
[d]Amsterdam Institute for Global Health and Development, Amsterdam, The Netherlands
[e]Amsterdam UMC, University of Amsterdam, Experimental Immunology, Amsterdam, The Netherlands
[f]Amsterdam UMC, University of Amsterdam, Medical Microbiology and Infection Prevention, Laboratory of Experimental Virology, Amsterdam, The Netherlands
[g]HIV Monitoring Foundation, Amsterdam, The Netherlands
[h]Public Health Service of Amsterdam, Infectious Diseases, Amsterdam, The Netherlands
[i]Amsterdam UMC, University of Amsterdam, Global Health, Amsterdam, The Netherlands

Myrthe Verburgh, Lisa van Pul and Marloes Grobben contributed equally to this article. Author order was determined by the respective number of hours each contributed to the project.
Marit van Gils, Neeltje Kootstra and Peter Reiss also contributed equally to this article.

Address correspondence to Myrthe L. Verburgh, m.l.verburgh@amsterdamumc.nl, Lisa van Pul, l.vanpul@amsterdamumc.nl, or Marloes Grobben, m.grobben@amsterdamumc.nl.

The authors declare a conflict of interest. FWNMW has served on scientific advisory boards for ViiV Healthcare and Gilead sciences. MFSvdL has received independent scientific grant support from Sanofi Pasteur, MSD Janssen Infectious Diseases and Vaccines, and Merck & Co; has served on advisory boards of GlaxoSmithKline and Merck & Co; and has received non-financial support from Stichting Pathologie Onderzoek en Ontwikkeling. MvdV through his institution has received independent scientific grant support and consultancy fees from AbbVie, Gilead Sciences, MSD, and ViiV Healthcare, for which honoraria were all paid to his institution. PR through his institution has received independent scientific grant support from Gilead Sciences, Janssen Pharmaceuticals Inc, Merck & Co and ViiV Healthcare, and has served on scientific advisory boards for Gilead Sciences, ViiV Healthcare, and Merck & Co honoraria for which were all paid to his institution. M.L.V., L.v.P., M.G., A.B., A.C.v.N., K.A.v.D., K.T., J.v.R., M.B., L.v.d.H., M.J.v.G., and N.A.K. declare no competing interests.

**ABSTRACT** Few studies have comprehensively compared severe acute respiratory syndrome coronavirus 2 (SARS-CoV-2) vaccine-induced and hybrid B- and T-cell responses in people with HIV (PWH) to those in comparable controls without HIV. We included 195 PWH and 246 comparable controls from the AGE<sub>h</sub>IV COVID-19 substudy. A positive nucleocapsid antibody (INgezim IgA/IgM/IgG) or self-reported PCR test defined prior SARS-CoV-2 infection. SARS-CoV-2 anti-spike (anti-S) IgG titers and anti-S IgG production by memory B cells were assessed. Neutralizing antibody titers were determined in a subset of participants. T-cell responses were assessed by gamma interferon (IFN-$\gamma$) release and activation-induced marker assay. We estimated mean differences in postvaccination immune responses ($\beta$) between levels of determinants. Anti-S IgG titers and anti-S IgG production by memory B cells were not different between PWH and controls. Prior SARS-CoV-2 infection ($\beta$ = 0.77), receiving mRNA vaccine ($\beta$ = 0.56), female sex ($\beta$ = 0.24), fewer days between last vaccination and sampling ($\beta$ = 0.07), and a CD4/CD8 ratio of <1.0 ($\beta$ = −0.39) were independently associated with anti-S IgG titers, but HIV status was not. Neutralization titers against the ancestral and Delta and Omicron SARS-CoV-2 variants were not different between PWH and controls. IFN-$\gamma$ release was higher in PWH. Prior SARS-CoV-2 infection ($\beta$ = 2.39), HIV-positive status ($\beta$ = 1.61), and fewer days between last vaccination and sampling ($\beta$ = 0.23) were independently associated with higher IFN-$\gamma$ release. The percentages of SARS-CoV-2-reactive CD4$^+$ and CD8$^+$ T cells, however, were not different between PWH and controls. Individuals with well-controlled HIV generally mount robust vaccine-induced as well as hybrid B- and T-cell immunity across SARS-CoV-2 vaccine platforms similar to controls. Determinants of a reduced vaccine response were likewise largely similar in both groups and included a lower CD4/CD8 ratio.

**IMPORTANCE** Some studies have suggested that people with HIV may respond less well to vaccines against SARS-CoV-2. We comprehensively compared B- and T-cell responses to different COVID-19 vaccines in middle-aged persons with well-treated HIV and individuals

of the same age without HIV, who were also highly comparable in terms of demographics and lifestyle, including those with prior SARS-CoV-2 infection. Individuals with HIV generally mounted equally robust immunity to the different vaccines. Even stronger immunity was observed in both groups after prior SARS-CoV-2 infection. These findings are reassuring with respect to the efficacy of SARS-Cov-2 vaccines for the sizable and increasing global population of people with HIV with access and a good response to HIV treatment.

**KEYWORDS** HIV, SARS-CoV-2 vaccines, humoral immune responses, cellular immune responses

Vaccination against severe acute respiratory syndrome coronavirus 2 (SARS-CoV-2) reduces the risk of symptomatic and severe COVID-19 (1, 2). Concerns have been raised about the immunogenicity of these vaccines in people with HIV (PWH). Studies have previously shown potential impaired immune responses in PWH to other vaccines or shorter duration of vaccine-induced protection (3).

Thus far, inconsistent findings have been reported on the short-term humoral immune response following SARS-CoV-2 vaccination in PWH. Some studies observed no significant differences in antibody titers against the SARS-CoV-2 spike (S) or receptor-binding domain (RBD) protein when comparing PWH to controls (4–6), while other studies reported significantly lower titers in PWH (7–10). Importantly, almost all of these studies have compared PWH to healthy controls who differed substantially with regard to age and/or sex (4–6, 8–10). Only Spinelli et al. have compared immune responses following vaccination between PWH on antiretroviral therapy (ART) (95% with suppressed HIV viremia) and controls who were matched by age, sex, type of mRNA vaccine, and days between vaccine administration and sampling, and they found lower anti-RBD IgG titers among PWH, particularly in those with lower CD4 counts (7).

Studies have shown that SARS-CoV-2-specific T-cell responses are essential for viral clearance, with a likely important role for long-lived memory T cells in protection against reinfection (11). Thus far, studies on cellular immune responses following SARS-CoV-2 vaccination in PWH have been limited, however, and found no significant differences between PWH and controls (5, 12).

Our aim was to compare both the humoral and cellular immune responses 4 to 13 weeks after completing primary SARS-CoV-2 vaccination in middle-aged individuals with well-controlled HIV and a group of individuals highly comparable in terms of demographics and lifestyle without HIV, participating in the ongoing AGE$_h$IV COVID-19 substudy in Amsterdam, The Netherlands (13).

## RESULTS

A total of 567 participants with an available prevaccination sample were included in the AGE$_h$IV COVID-19 substudy (Fig. 1). Postvaccination serum was obtained 4 to 13 weeks after the last vaccine dose for 441 participants. Most participants were Caucasian males (92.3% PWH and 83.3% controls), and the median age was 62 years (63 in PWH and 62 in controls) (Table 1). PWH were known to have had HIV for a median 22.6 years, and their median nadir CD4 count was 180/mm$^3$. All PWH were on ART (99.5% virologically suppressed), and their current median CD4 count was 640/mm$^3$. Prior SARS-CoV-2 infection was detected in 28 PWH (14.4%) and 34 controls (13.8%), of whom 8 and 9, respectively, acquired SARS-CoV-2 infection between the pre- and postvaccination samplings.

The vaccines received were BNT162b2 ($n$ = 289 [65.5%]), mRNA-1273 ($n$ = 14 [3.2%]), ChAdOx1 ($n$ = 125 [28.3%]), Ad26.COV2.S ($n$ = 9 [2.0%]), or heterologous ChAdOx1 plus BNT162b2 ($n$ = 4 [0.9%]), distributed similarly between PWH and controls. Ten participants received only a single dose of BNT162b2, mRNA-1273, or ChAdOx1 because of prior SARS-CoV-2 infection.

**SARS-CoV-2 humoral immune responses.** To assess whether PWH and controls had comparable humoral immune responses, SARS-CoV-2 anti-S IgG and anti-RBD IgG was measured in all 441 participants. Prevaccination anti-S IgG titers were not significantly different between PWH and controls (Fig. 2A). Postvaccination anti-S IgG titers

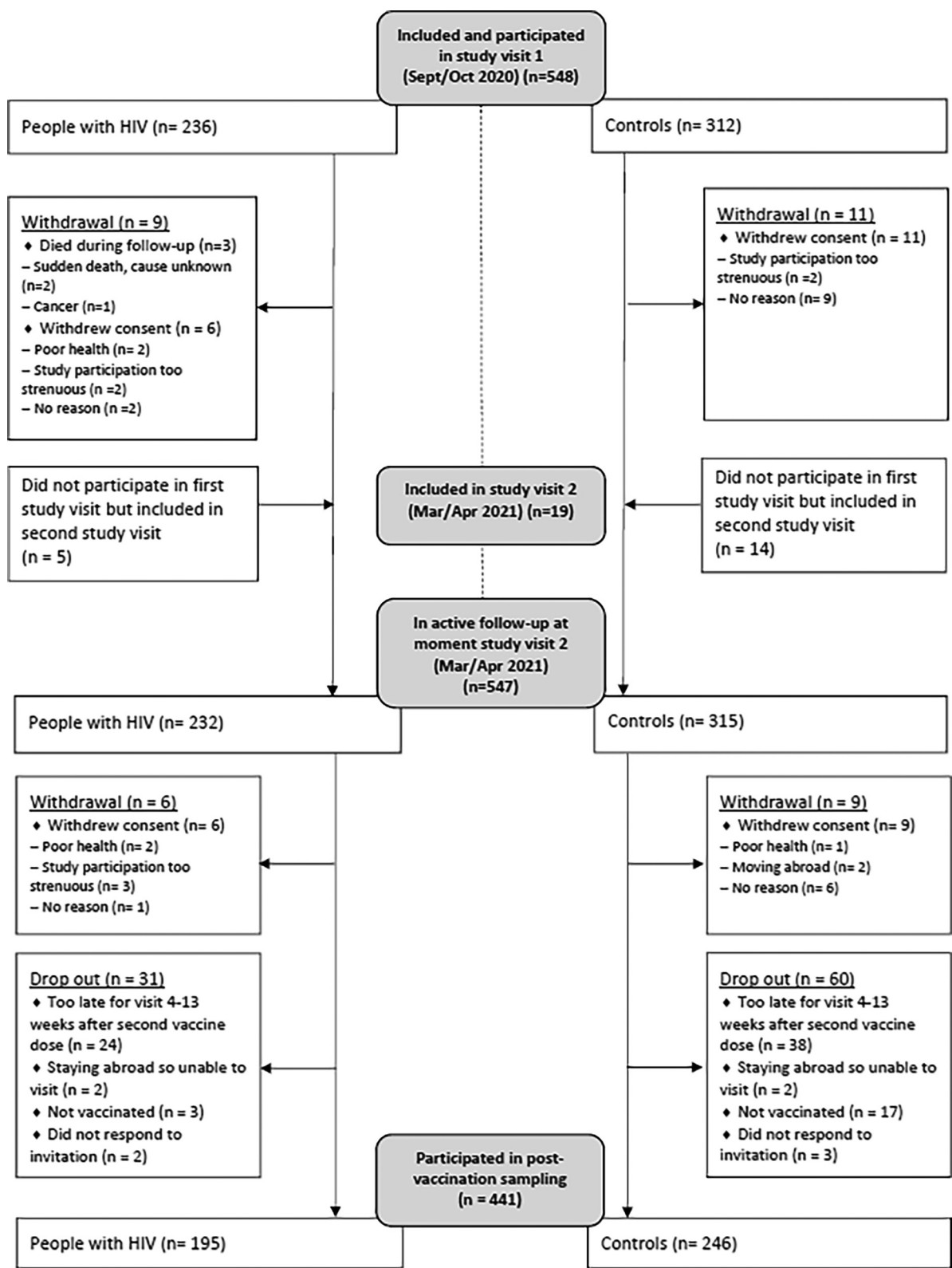

**FIG 1** Overview of inclusion in the AGE$_h$IV COVID-19 substudy (September 2020 to April 2021) and in analyses of postvaccination immune responses.

**TABLE 1** Characteristics by HIV status of 441 participants included in the AGE$_h$IV COVID-19 substudy with a sample available at 4 to 13 weeks after the last dose of a SARS-CoV-2 vaccine[a]

| Parameter | Result for: | | P value[b] |
| --- | --- | --- | --- |
| | PWH (n = 195) | Controls (n = 246) | |
| Age, yrs, median (IQR)[c] | 63.3 (58.7–68.3) | 61.6 (58.0–67.5) | 0.145* |
| Age group, yrs, n (%) | | | 0.393** |
| <60 | 66 (33.9) | 101 (41.0) | |
| 60–64 | 49 (25.1) | 59 (24.0) | |
| 65–69 | 41 (21.0) | 40 (16.3) | |
| ≥70 | 39 (20.0) | 46 (18.7) | |
| Male sex at birth, n (%) | 184 (94.4) | 211 (85.8) | 0.003** |
| Ethnic origin, n (%) | | | 0.109*** |
| Caucasian | 190 (97.4) | 237 (96.4) | |
| African | 5 (2.6) | 4 (1.6) | |
| Asian | 0 (0.0) | 5 (2.0) | |
| Educational level, n (%)[d] | | | 0.061*** |
| Lower education (primary or secondary) | 91 (47.1) | 89 (37.1) | |
| Higher vocational or university education | 97 (50.3) | 147 (61.2) | |
| Other | 5 (2.6) | 4 (1.7) | |
| BMI, kg/m$^2$, median (IQR)[e] | 24.7 (23.0–27.3) | 25.0 (23.2–27.5) | 0.391* |
| BMI group, n (%) | | | |
| Underweight (<18.5 kg/m$^2$) | 1 (0.5) | 0 (0.0) | 0.608*** |
| Normal weight (18.5–24.9 kg/m$^2$) | 101 (51.8) | 118 (48.0) | |
| Overweight (25.0–29.9 kg/m$^2$) | 72 (36.9) | 97 (39.4) | |
| Obese (≥30.0 kg/m$^2$) | 21 (10.8) | 31 (12.6) | |
| Total comorbidities, n (%)[e] | | | <0.001** |
| 0 | 79 (40.5) | 154 (62.6) | |
| 1–2 | 96 (49.2) | 78 (31.7) | |
| 3–7 | 20 (10.3) | 14 (5.7) | |
| Current CD4/CD8 ratio, median (IQR)[e] | 0.86 (0.65–1.22) | 1.87 (1.32–2.56) | <0.001* |
| CD4/CD8 ratio group, n (%) | | | |
| <0.50 | 21 (10.8) | 0 (0.0) | <0.001*** |
| 0.50–0.99 | 98 (50.2) | 21 (8.5) | |
| ≥1.0 | 76 (39.0) | 225 (91.5) | |
| Current CD4 count, cells/mm$^3$, median (IQR)[e] | 640 (500–850) | 810 (650–1,010) | <0.001* |
| CD4 count (cells/mm$^3$) group, n (%) | | | |
| <350 | 19 (9.8) | 3 (1.2) | <0.001*** |
| 350–499 | 26 (13.3) | 21 (8.5) | |
| 500–749 | 77 (39.5) | 72 (29.3) | |
| ≥750 | 73 (37.4) | 150 (61.0) | |
| Current CD8 count, cells/mm$^3$, median (IQR)[e] | 750 (500–990) | 410 (300–560) | <0.001* |
| CD8 count (cells/mm$^3$) group, n (%) | | | |
| <350 | 22 (11.3) | 88 (35.8) | <0.001*** |
| 350–499 | 24 (12.3) | 73 (29.6) | |
| 500–749 | 50 (25.6) | 42 (17.1) | |
| ≥750 | 99 (50.8) | 43 (17.5) | |
| Time since HIV diagnosis, yrs, median (IQR)[c] | 22.6 (17.1–27.9) | NA | NA |
| Time since first starting ART, yrs, median (IQR)[c] | 19.8 (13.9–24.7) | NA | NA |
| CD4 nadir, cells/mm$^3$, median (IQR) | 180 (70–260) | NA | NA |

**TABLE 1** (Continued)

| Parameter | Result for: | | P value[b] |
|---|---|---|---|
| | PWH (n = 195) | Controls (n = 246) | |
| Undetectable HIV-1 viral load, n (%)[e,f] | 193 (99.5) | NA | NA |
| SARS-CoV-2 vaccine type, n (%) | | | 0.491*** |
| BNT162b2 | 122 (62.6) | 167 (67.9) | |
| mRNA-1273 | 8 (4.1) | 6 (2.4) | |
| ChAdOx1 | 61 (31.3) | 64 (26.0) | |
| Ad26.COV2.S | 3 (1.5) | 6 (2.4) | |
| ChAdOx1 + BNT162b2 | 1 (0.5) | 3 (1.2) | |
| Only 1 dose of BNT162b2, mRNA-1273, or ChAdOx1 due to prior SARS-CoV-2 infection, n (%) | 2 (1.0) | 8 (3.3) | 0.284*** |
| Days between prevaccination sample and first vaccine dose, median (IQR)[g] | 44 (26–67) | 45 (28–74) | 0.702* |
| Days between 1st and 2nd vaccine doses, median (IQR)[h] | | | |
| BNT162b2 | 35 (35–36) | 36 (35–36) | 0.096* |
| mRNA-1273 | 28 (28–32) | 32 (28–36) | 0.459* |
| ChAdOx1 | 77 (63–77) | 76 (68–77) | 0.538* |
| ChAdOx1 + BNT162b2 | 88 (88–88) | 65 (33–113) | 0.655* |
| Days between last vaccine dose and postvaccination sample, median (IQR)[i] | 64 (46–76) | 70 (43–77) | 0.262* |
| Prior SARS-CoV-2 infection, n (%) | | | 0.877** |
| Prior to prevaccination sample | 20 (10.3) | 25 (10.2) | |
| Between pre- and postvaccination sample | 8 (4.1) | 9 (3.7) | |

[a]General abbreviations: PWH, people with HIV; IQR, interquartile range; ART, antiretroviral therapy; BMI, body mass index; NA, not applicable.
[b]Significance determination: *, Wilcoxon rank sum test; **, Pearson $\chi^2$ test; ***, Fisher's exact test.
[c]At the moment of postvaccination blood draw.
[d]Missing in 2/195 PWH and 6/246 controls.
[e]Last available data prior to receiving first vaccine dose of the primary vaccination course.
[f]HIV viral load was missing in 1/195 PWH.
[g]The first vaccine dose was either (i) the first dose of a two-dose vaccine regimen, (ii) one dose of Ad26.COV2.S, or (iii) one dose of BNT162b2, mRNA-1273, or ChAdOx1 in those who received only one vaccine dose.
[h]In 422/441 participants with a two-vaccine-dose regimen.
[i]The last vaccine dose was either (i) the second dose of BNT162b2, mRNA-1273, or ChAdOx1, (ii) one dose of Ad26.COV2.S, or (iii) one dose of BNT162b2, mRNA-1273, or ChAdOx1 in those who received only one vaccine dose.

in both groups differed by vaccine type, with higher titers in mRNA-based vaccine recipients ($P < 0.001$ versus vector based) (Fig. 2B). Postvaccination anti-S IgG titers were higher in participants with prior SARS-CoV-2 infection ($P < 0.001$ versus without prior SARS-CoV-2 infection), but also not significantly different between PWH and controls. Results for anti-RBD IgG titers were highly comparable (see Fig. S1 in the supplemental material).

Six PWH (3.1%) and six controls (2.4%) showed no response in SARS-CoV-2 anti-S IgG titer after vaccination (titer of <17.8 mean fluorescent intensity [MFI]) (Fig. 2B). All received a vector-based vaccine (11 received ChAdOx1, and 1 received Ad26.COV2.S) and had significantly more days elapsed between their last vaccine dose and postvaccination sampling date than in responders (median of 80 versus 65 days) (see Table S1 in the supplemental material for details).

Multivariable analyses demonstrated that HIV status was not independently associated with anti-S IgG titers, whereas prior SARS-CoV-2 infection, receipt of a mRNA vaccine, and female sex were each independently associated with higher anti-S IgG titers (Table 2). A greater number of days between last vaccine dose and moment of sampling and a current CD4/CD8 ratio of <1.0 were each associated with lower titers. When stratified by HIV status, a current CD4/CD8 ratio of <1.0 remained associated with lower anti-S IgG titers in both PWH (Table S2) and controls (Table S3). Factors associated with postvaccination anti-RBD IgG titers were comparable to those associated with anti-S IgG titers (Tables S4 to S6).

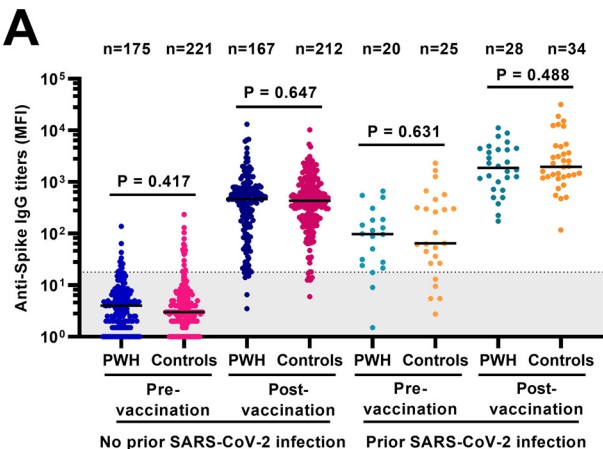

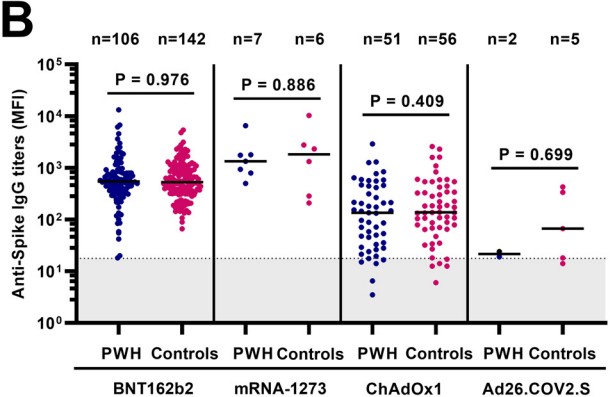

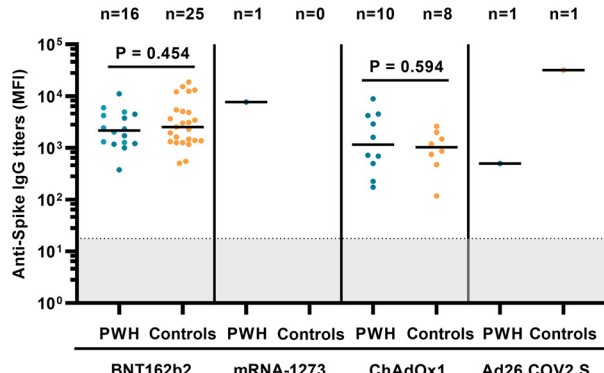

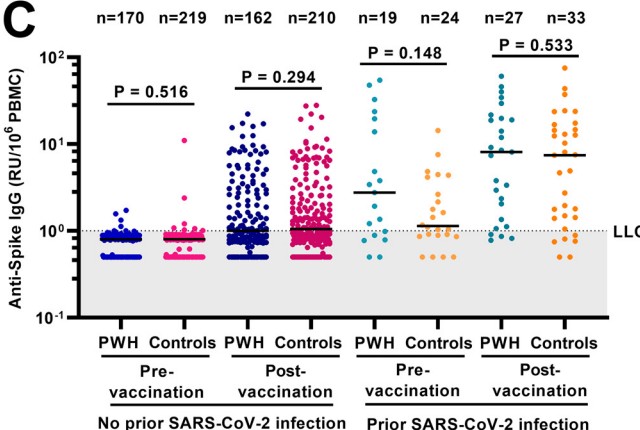

**FIG 2** SARS-CoV-2 anti-spike IgG titers and anti-spike IgG production by memory B cells pre- and postvaccination against SARS-CoV-2 in 441 participants of the AGE$_h$IV COVID-19 substudy by HIV status. (A) Pre- and postvaccination anti-S IgG titers stratified by prior SARS-CoV-2 infection and HIV status; (B) postvaccination anti-S IgG titers in those without (left panel) and with (right panel) prior SARS-CoV-2 infection stratified by vaccine type; (C) anti-S IgG production by memory B cells upon *in vitro* stimulation of PBMCs (the dotted line indicates the lower limit of quantification [LLQ]) (LLQ = 1) in 432 of 441 participants. Results are expressed as the median fluorescence intensity (MFI) of at least 50 beads per antigen. The dotted line represents the antibody nonresponse cutoff value (anti-spike IgG titer of <17.8 MFI). P values, comparing people with HIV (PWH) and controls, were calculated using the Wilcoxon rank sum test.

Anti-S IgG production by memory B cells upon *in vitro* stimulation of peripheral blood mononuclear cells (PBMCs) was measured in 432 participants, and the levels were not significantly different between PWH and controls (Fig. 2C). Memory B-cell response was detectable postvaccination in 157 PWH (83.1%) and 219 controls (90.1%).

In the subgroup of 40 PWH and 40 controls (whose characteristics are summarized in Table 3), neutralizing antibodies against the ancestral and Delta variants, but not against the

**TABLE 2** Factors associated with SARS-CoV-2 anti-spike IgG titers following vaccination against SARS-CoV-2 in 441 participants in the AGE$_h$IV COVID-19 substudy[a]

| Parameter | n (%)[b] | Univariable analysis | | Multivariable analysis | |
|---|---|---|---|---|---|
| | | Log$_{10}$ MFI, $\beta$ (95% CI)[c] | P value | Log$_{10}$ MFI, $\beta$ (95% CI)[c] | P value |
| HIV status | | | 0.275 | | 0.190 |
| Negative | 246 (55.8) | REF | | REF | |
| Positive | 195 (44.2) | −0.07 (−0.18 to +0.05) | | +0.07 (−0.03 to +0.17) | |
| Age, yrs[d] | | | <0.001 | NI | |
| <60 | 167 (37.9) | REF | | | |
| 60–64 | 108 (24.5) | −0.36 (−0.51 to −0.22) | | | |
| 65–69 | 81 (18.3) | −0.23 (−0.39 to −0.07) | | | |
| ≥70 | 85 (19.3) | −0.04 (−0.20 to +0.12) | | | |
| Sex at birth | | | 0.001 | | 0.001 |
| Male | 395 (89.6) | REF | | REF | |
| Female | 46 (10.4) | +0.31 (+0.12 to +0.50) | | +0.24 (+0.10 to +0.39) | |
| Ethnic origin | | | 0.041 | NI | |
| Caucasian | 427 (96.8) | REF | | | |
| African | 9 (2.1) | +0.52 (+0.11 to +0.93) | | | |
| Asian | 5 (1.1) | +0.16 (−0.39 to +0.71) | | | |
| BMI group[e] | | | 0.316 | NI | |
| Underweight (<18.5 kg/m$^2$) | 1 (0.2) | −0.14 (−1.37 to +1.09) | | | |
| Normal weight (18.5–24.9 kg/m$^2$) | 219 (49.7) | REF | | | |
| Overweight (25.0–29.9 kg/m$^2$) | 169 (38.3) | +0.10 (−0.02 to +0.23) | | | |
| Obese (≥30.0 kg/m$^2$) | 52 (11.8) | −0.05 (−0.23 to +0.14) | | | |
| Total comorbidities[e] | | | 0.762 | NI | |
| 0 | 233 (52.8) | REF | | | |
| 1–2 | 174 (39.5) | −0.05 (−0.17 to +0.08) | | | |
| 3–7 | 34 (7.7) | −0.02 (−0.24 to +0.21) | | | |
| Prior SARS-CoV-2 infection | | | <0.001 | | <0.001 |
| No | 379 (85.9) | REF | | REF | |
| Yes | 62 (14.1) | +0.75 (+0.60 to +0.91) | | +0.77 (+0.65 to +0.90) | |
| SARS-CoV-2 vaccine | | | <0.001 | | <0.001 |
| mRNA-based | 303 (68.7) | +0.61 (+0.50 to +0.72) | | +0.56 (+0.46 to +0.65) | |
| Vector-based | 134 (30.4) | REF | | REF | |
| Heterologous[f] | 4 (0.9) | +0.55 (−0.009 to +1.10) | | +0.53 (+0.07 to +0.99) | |
| Days between last vaccine dose and postvaccination sampling (per 10-day increase) | | −0.10 (−0.13 to −0.07) | <0.001 | −0.07 (−0.10 to −0.05) | <0.001 |
| Current CD4/CD8 ratio[e] | | | 0.006 | | <0.001 |
| <0.50 | 21 (4.8) | −0.37 (−0.65 to −0.10) | | −0.47 (−0.69 to −0.25) | |
| 0.50–0.99 | 119 (27.0) | −0.14 (−0.27 to −0.01) | | −0.13 (−0.24 to −0.01) | |
| ≥1.0 | 301 (68.3) | REF | | REF | |
| Current CD4 count, cells/mm$^{3e}$ | | | 0.091 | NI | |
| <350 | 22 (5.0) | −0.20 (−0.47 to +0.07) | | | |
| 350–499 | 47 (10.7) | −0.18 (−0.37 to +0.02) | | | |
| 500–749 | 149 (33.8) | −0.13 (−0.26 to +0.0005) | | | |
| ≥750 | 223 (50.6) | REF | | | |
| Current CD8 count, cells/mm$^{3e}$ | | | 0.201 | NI | |
| <350 | 110 (24.9) | REF | | | |
| 350–499 | 97 (22.0) | −0.05 (−0.22 to +0.12) | | | |
| 500–749 | 92 (20.9) | +0.01 (−0.16 to +0.18) | | | |
| ≥750 | 142 (32.2) | −0.14 (−0.30 to +0.01) | | | |

[a]General abbreviations: BMI, body mass index; CI, confidence interval; REF, reference group; NI, variable not included in multivariable analysis.
[b]Number (total n = 441) and percentage of participants for each variable category.
[c]Values represent regression coefficients ($\beta$) of linear regression with 95% CI. The unit is the log$_{10}$ median fluorescence intensity (MFI).
[d]At the moment of postvaccination blood draw.
[e]The last available data prior to receiving first vaccine dose of the primary vaccination course.
[f]Received one dose of ChAdOx1 and one dose of BNT162b2.

Omicron BA.1 variant, could be detected. There were no significant differences in neutralization titers between PWH and controls against either the ancestral or Delta variant (Fig. 3).

Multivariable analyses showed that HIV status was not independently associated with neutralization titers against ancestral SARS-CoV-2, but age of <60 years (versus

**TABLE 3** Characteristics by HIV status of 80 participants included in a subgroup analysis in the AGE$_h$IV COVID-19 substudy (at 4 to 13 weeks after last dose of a SARS-CoV-2 vaccine)[a]

| Parameter | Result for: | | P value[b] |
|---|---|---|---|
| | PWH (n = 40) | Controls (n = 40) | |
| Age, yrs, median (IQR)[c] | 63.9 (58.8–69.5) | 63.4 (58.8–69.6) | NS |
| Age group, yrs, n (%) | | | |
| <60 | 14 (35.0) | 12 (30.0) | |
| 60–64 | 6 (15.0) | 11 (27.5) | |
| 65–69 | 12 (30.0) | 8 (20.0) | |
| ≥70 | 8 (20.0) | 9 (22.5) | |
| Male sex at birth, n (%) | 38 (95.0) | 38 (95.0) | NS |
| Ethnic origin, n (%) | | | >0.999*** |
| Caucasian | 39 (97.5) | 38 (95.0) | |
| African | 1 (2.5) | 1 (2.5) | |
| Asian | 0 (0.0) | 1 (2.5) | |
| BMI, kg/m$^2$, median (IQR)[d] | 24.4 (22.5–26.9) | 24.6 (23.5–27.7) | 0.394* |
| BMI group, n (%) | | | |
| Underweight (<18.5 kg/m$^2$) | 0 (0.0) | 0 (0.0) | 0.756** |
| Normal weight (18.5–24.9 kg/m$^2$) | 24 (60.0) | 21 (52.5) | |
| Overweight (25.0–29.9 kg/m$^2$) | 11 (27.5) | 14 (35.0) | |
| Obese (≥30.0 kg/m$^2$) | 5 (12.5) | 5 (12.5) | |
| Total comorbidities, n (%)[d] | | | 0.219*** |
| 0 | 16 (40.0) | 24 (60.0) | |
| 1–2 | 20 (50.0) | 13 (32.5) | |
| 3–7 | 4 (10.0) | 3 (7.5) | |
| Current CD4/8 ratio, median (IQR)[d] | 0.79 (0.57–1.16) | 1.68 (1.15–2.29) | <0.001* |
| CD4/CD8 ratio group, n (%) | | | |
| <0.50 | 6 (15.0) | 0 (0.0) | <0.001*** |
| 0.50–0.99 | 21 (52.5) | 7 (17.5) | |
| ≥1.0 | 13 (32.5) | 33 (82.5) | |
| Current CD4 count, cells/mm$^3$, median (IQR)[d] | 690 (515–930) | 795 (585–950) | 0.231* |
| CD4 count (cells/mm$^3$) group, n (%) | | | |
| <350 | 3 (7.5) | 0 (0.0) | 0.316*** |
| 350–499 | 7 (17.5) | 5 (12.5) | |
| 500–749 | 13 (32.5) | 13 (32.5) | |
| ≥750 | 17 (42.5) | 22 (55.0) | |
| Current CD8 count, cells/mm$^{3d}$ | 790 (590–1155) | 445 (335–625) | <0.001* |
| CD8 count (cells/mm$^3$) group, n (%) | | | |
| <350 | 5 (12.5) | 11 (27.5) | <0.001*** |
| 350–499 | 1 (2.5) | 13 (32.5) | |
| 500–749 | 11 (27.5) | 9 (22.5) | |
| ≥750 | 23 (57.5) | 7 (17.5) | |
| Time since HIV diagnosis, yrs, median (IQR)[c] | 23.7 (20.0–28.7) | NA | NA |
| Time since first starting ART, yrs, median (IQR)[c] | 22.0 (17.9–25.7) | NA | NA |
| CD4 nadir, cells/mm$^3$ | 155 (40–265) | NA | NA |
| Undetectable HIV-1 viral load, n (%)[d] | 40 (100) | NA | NA |

**TABLE 3** (Continued)

| Parameter | Result for: | | P value[b] |
| | PWH (*n* = 40) | Controls (*n* = 40) | |
|---|---|---|---|
| SARS-CoV-2 vaccine type | | | NS |
| BNT162b2 | 25 (62.5) | 25 (62.5) | |
| mRNA-1273 | 4 (10.0) | 4 (10.0) | |
| ChAdOx1 | 11 (27.5) | 11 (27.5) | |
| Only 1 dose of BNT162b2, mRNA-1273, or ChAdOx1 due to prior SARS-CoV-2 infection, *n* (%) | 0 (0.0) | 1 (2.5) | >0.999*** |
| Days between prevaccination sample and 1st vaccine dose, median (IQR)[e] | 50 (32–91) | 42 (26–79) | 0.427* |
| Days between last vaccine dose and postvaccination sample, median (IQR)[f] | 65 (48–78) | 72 (45–78) | 0.711* |

[a]General abbreviations: PWH, people with HIV; IQR, interquartile range; ART, antiretroviral therapy; BMI, body mass index; NA, not applicable.
[b]Significance determination: *, Wilcoxon rank sum test; **, Pearson $\chi^2$ test, ***, Fisher's exact test. NS, not shown (*P* values for age, sex, and vaccine type omitted because of matching on these variables).
[c]At the moment of the postvaccination blood draw.
[d]The last available data prior to receiving first vaccine dose of the primary vaccination course.
[e]The first vaccine dose was either (i) the first dose of a two-dose vaccine regimen, (ii) one dose of Ad26.COV2.S, or (iii) one dose of BNT162b2, mRNA-1273, or ChAdOx1 in those who received only one vaccine dose.
[f]The last vaccine dose was either (i) the second dose of BNT162b2, mRNA-1273, or ChAdOx1, (ii) one dose of Ad26.COV2.S, or (iii) one dose of BNT162b2, mRNA-1273, or ChAdOx1 in those who received only one vaccine dose.

≥70 years) and receiving mRNA-1273 (versus BNT162b2) were independently associated with higher titers (Table 4). A greater number of days between last vaccine dose and moment of sampling as well as a current CD4/CD8 ratio of <1.0 were associated with lower titers.

For the Delta variant, only receiving the mRNA-1273 vaccine was associated with higher neutralization titers ($\beta$, +0.34 $\log_{10}$ 50% infective dose [$ID_{50}$], with a 95% confidence interval [CI] of 0.12 to 0.56, compared to BNT162b2).

**SARS-CoV-2 cellular immune responses.** The SARS-CoV-2-specific T-cell response by gamma interferon (IFN-$\gamma$) release assay was assessed in 436 participants and was detectable in 109 PWH (56.8%) and 79 controls (32.4%) postvaccination. Among 392 participants without prior SARS-CoV-2 infection, prevaccination IFN-$\gamma$ release was significantly higher in PWH than controls, but not among 44 participants with prior SARS-CoV-2 infection (Fig. 4A). Postvaccination IFN-$\gamma$ release was also higher in PWH, only reaching statistical significance in those without prior SARS-CoV-2 infection and vaccinated with BNT162b2 or ChAdOx1 (Fig. 4B).

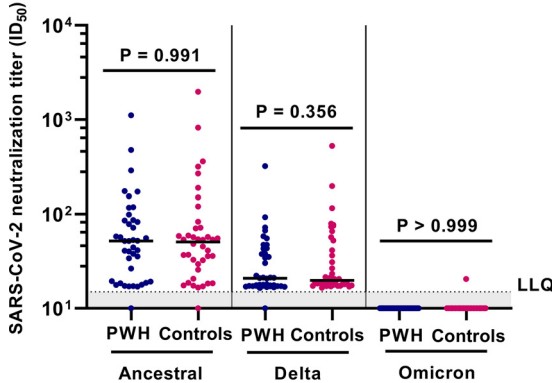

**FIG 3** Neutralization against SARS-CoV-2 ancestral, Delta, and Omicron BA.1 variants after SARS-CoV-2 vaccination in 80 participants of the AGE$_h$IV COVID-19 substudy by HIV status. People with HIV (PWH; *n* = 40) and controls (*n* = 40) were 1:1 matched on age, sex at birth, and vaccine type. Results are expressed as the serum dilution at which 50% of infectivity of SARS-CoV-2 (ancestral, Delta, and Omicron variants) was inhibited ($ID_{50}$). The dotted line indicates the lower limit of quantification (LLQ) (LLQ = 15). *P* values, comparing PWH and controls, were calculated using Wilcoxon rank sum test.

**TABLE 4** Factors associated with neutralization against ancestral SARS-CoV-2 following vaccination against SARS-CoV-2 in 80 participants in the AGE$_h$IV COVID-19 substudy[a]

| Parameter | n (%)[b] | Univariable analysis | | Multivariable analysis | |
|---|---|---|---|---|---|
| | | Log$_{10}$ ID$_{50}$, $\beta$ (95% CI)[c] | P value | Log$_{10}$ ID$_{50}$, $\beta$ (95% CI)[c] | P value |
| HIV status | | | 0.819 | | 0.234 |
| Negative | 40 (50.0) | REF | | REF | |
| Positive | 40 (50.0) | −0.02 (−0.21 to +0.17) | | +0.12 (−0.08 to +0.31) | |
| Age, yrs[d] | | | 0.018 | | 0.022 |
| <60 | 26 (32.5) | REF | | REF | |
| 60–64 | 17 (21.3) | −0.25 (−0.51 to +0.007) | | −0.13 (−0.42 to +0.15) | |
| 65–69 | 20 (25.0) | −0.35 (−0.60 to −0.11) | | −0.31 (−0.53 to −0.09) | |
| ≥70 | 17 (21.3) | −0.32 (−0.58 to −0.07) | | −0.30 (−0.54 to −0.06) | |
| Sex at birth | | | 0.280 | NI | |
| Male | 76 (95.0) | REF | | | |
| Female | 4 (5.0) | +0.25 (−0.20 to +0.71) | | | |
| Ethnic origin | | | 0.530 | NI | |
| Caucasian | 77 (96.2) | REF | | | |
| African | 2 (2.5) | +0.36 (−0.27 to +0.98) | | | |
| Asian | 1 (1.3) | +0.03 (−0.85 to +0.90) | | | |
| BMI group[e] | | | 0.596 | NI | |
| Underweight (<18.5 kg/m$^2$) | 0 (0.0) | | | | |
| Normal weight (18.5–24.9 kg/m$^2$) | 44 (55.0) | REF | | | |
| Overweight (25.0–29.9 kg/m$^2$) | 26 (32.5) | +0.03 (−0.19 to +0.25) | | | |
| Obese (≥30.0 kg/m$^2$) | 10 (12.5) | −0.14 (−0.44 to +0.17) | | | |
| Total comorbidities[e] | | | 0.566 | NI | |
| 0 | 40 (50.0) | REF | | | |
| 1–2 | 33 (41.3) | −0.01 (−0.22 to +0.19) | | | |
| 3–7 | 7 (8.7) | −0.19 (−0.55 to +0.17) | | | |
| SARS-CoV-2 vaccine type | | | <0.001 | | 0.006 |
| BNT162b2 | 50 (62.5) | REF | | REF | |
| mRNA-1273 | 8 (10.0) | +0.54 (+0.23 to +0.84) | | +0.43 (+0.13 to +0.73) | |
| ChAdOx1 | 22 (27.5) | −0.11 (−0.32 to +0.10) | | −0.06 (−0.30 to +0.19) | |
| Days between last vaccine dose and postvaccination sampling (per 10-day increase) | | −0.06 (−0.12 to −0.01) | 0.015 | −0.06 (−0.10 to −0.009) | 0.019 |
| Current CD4/CD8 ratio[e] | | | 0.108 | | 0.014 |
| <0.50 | 6 (7.5) | −0.26 (−0.63 to +0.11) | | −0.27 (−0.62 to +0.08) | |
| 0.50–0.99 | 28 (35.0) | −0.19 (−0.39 to +0.01) | | −0.29 (−0.49 to −0.09) | |
| ≥1.0 | 46 (57.5) | REF | | REF | |
| Current CD4 count, cells/mm$^3$[e] | | | 0.369 | NI | |
| <350 | 3 (3.8) | +0.13 (−0.39 to +0.65) | | | |
| 350–499 | 12 (15.0) | −0.23 (−0.52 to +0.06) | | | |
| 500–749 | 26 (32.5) | +0.0006 (−0.22 to +0.22) | | | |
| ≥750 cells | 39 (48.7) | REF | | | |
| Current CD8 count, cells/mm$^3$[e] | | | 0.319 | NI | |
| <350 | 16 (20.0) | REF | | | |
| 350–499 | 14 (17.5) | −0.25 (−0.56 to +0.06) | | | |
| 500–749 | 20 (25.0) | −0.07 (−0.36 to +0.21) | | | |
| ≥750 | 30 (37.5) | −0.20 (−0.47 to +0.07) | | | |

[a]General abbreviations: CI, confidence interval; REF, reference group; NI, variable not included in multivariable analysis.
[b]Number (total n = 80) and percentage of participants for each variable category.
[c]Values represent regression coefficients ($\beta$) and their 95% CI of linear regression, with a random intercept to account for variation between pairs. The unit is the log$_{10}$ serum dilution at which 50% of the infectivity was inhibited (ID$_{50}$).
[d]At the moment of postvaccination blood draw.
[e]The last available data prior to receiving first vaccine dose of the primary vaccination course.

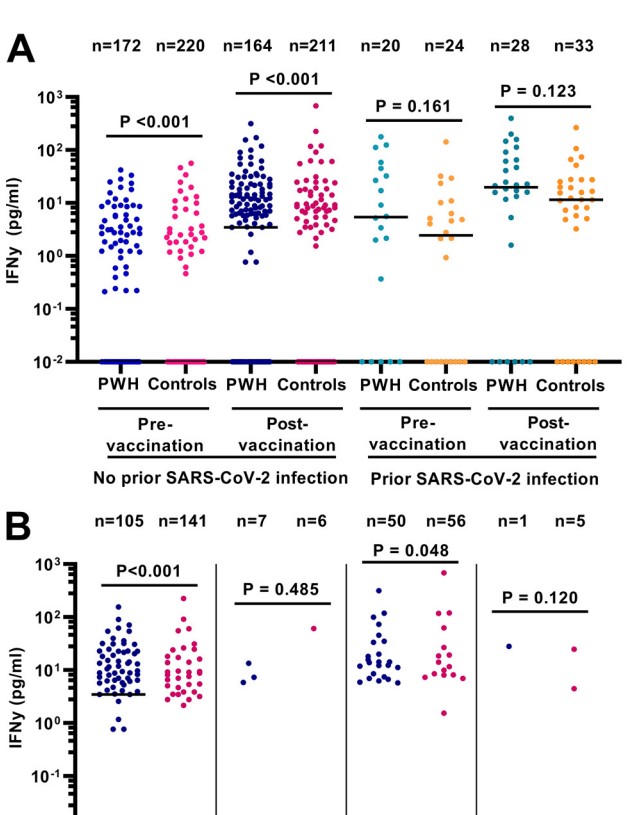

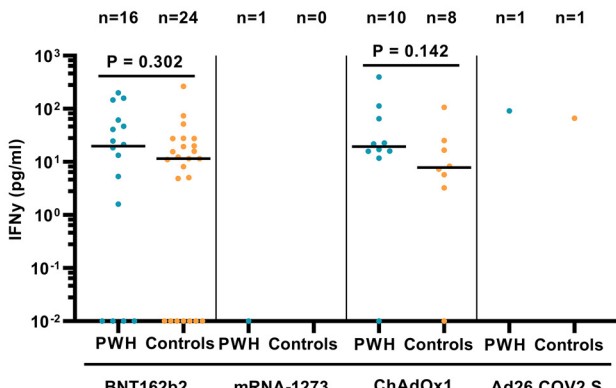

**FIG 4** SARS-CoV-2-specific IFN-γ release pre- and postvaccination against SARS-CoV-2 in 436 participants of the AGE_hIV COVID-19 substudy by HIV status. SARS-CoV-2 T-cell responses were measured by IFN-γ release assay by PBMCs upon SARS-CoV-2 peptide stimulation from 436 of 441 participants. (A) Pre- and postvaccination IFN-γ release stratified by prior SARS-CoV-2 infection and HIV status; (B) postvaccination IFN-γ release in those without (left panel) and with (right panel) prior SARS-CoV-2 infection stratified by vaccine type. Results are expressed as the release of IFN-γ (picograms per milliliter). *P* values, comparing people with HIV (PWH) and controls, were calculated using the Wilcoxon rank sum test.

Multivariable analysis with an adjustment for prevaccination IFN-γ release showed that HIV-positive status and prior SARS-CoV-2 infection were independently associated with higher IFN-γ release (Table 5). A greater number of days between last vaccine dose and moment of sampling was independently associated with lower IFN-γ release. The observed responses were not significantly associated with vaccine type, sex, and CD4/CD8 ratio in all participants. When stratified by HIV status, however, a current CD4/CD8 ratio of <1.0 was associated with lower IFN-γ release in PWH (Table S7), but not in controls (Table S8).

The percentages of SARS-CoV-2 antigen-reactive CD4$^+$ and CD8$^+$ T cells were determined in the subgroup of 40 PWH and 40 controls with available PBMC samples (*n* = 36 and *n* = 35, respectively) by the activation-induced marker (AIM) assay. The characteristics of these 71 participants were similar to the complete subset of 80 participants (data not shown). The percentages of both reactive CD4$^+$ and CD8$^+$ T cells were not statistically significantly different between PWH and controls, with a tendency toward a lower percentage of reactive CD4$^+$ T cells in PWH (Fig. 5).

**Immune correlates of SARS-CoV-2-specific immune responses.** To determine immune correlates associated with the vaccine response, T-cell and monocyte phenotyping was performed in the subgroup of 36 PWH and 35 controls by flow cytometry (Table S9). After adjusting for the factors found to be associated with the SARS-CoV-2-specific T-cell response by IFN-γ release assay in the overall study population, IFN-γ release showed a significantly positive association with CD163 expression on classical monocytes (Table S10). The percentage of reactive CD4$^+$ T cells was significantly negatively associated with the expression of CD163 on classical monocytes and positively associated with CD163 and CD64 on the CD16$^+$ monocytes (Table S11). The percentage of reactive CD8$^+$ T cells was significantly negatively

**TABLE 5** Factors associated with SARS-CoV-2 IFN-$\gamma$ release following vaccination against SARS-CoV-2 in 436 participants in the AGE$_h$IV COVID-19 substudy[a]

| Parameter | n (%)[b] | Univariable analysis[c] Log$_{10}$ pg/mL, $\beta$ (95% CI)[d] | P value | Multivariable analysis[c] Log$_{10}$ pg/mL, $\beta$ (95% CI)[d] | P value |
|---|---|---|---|---|---|
| HIV status | | | <0.001 | | <0.001 |
| Negative | 244 (56.0) | REF | | REF | |
| Positive | 192 (44.0) | +1.59 (+0.91 to +2.27) | | +1.61 (+0.96 to +2.26) | |
| | | | | | |
| Age, yrs[e] | | | 0.256 | NI | |
| <60 | 164 (37.6) | REF | | | |
| 60–64 | 106 (24.3) | +0.54 (−0.34 to +1.42) | | | |
| 65–69 | 81 (18.6) | +0.52 (−0.44 to +1.48) | | | |
| ≥70 | 85 (19.5) | −0.38 (−1.38 to +0.63) | | | |
| | | | | | |
| Sex at birth | | | 0.712 | NI | |
| Male | 391 (89.7) | REF | | | |
| Female | 45 (10.3) | +0.21 (−0.90 to +1.32) | | | |
| | | | | | |
| Ethnic origin | | | 0.436 | NI | |
| Caucasian | 422 (96.8) | REF | | | |
| African | 9 (2.1) | +1.33 (−0.92 to +3.58) | | | |
| Asian | 5 (1.1) | +0.91 (−2.13 to +3.96) | | | |
| | | | | | |
| BMI group[f] | | | 0.439 | NI | |
| Underweight (<18.5) | 1 (0.2) | −14.67 (−1138.98 to +1109.632) | | | |
| Normal weight (18.5–24.9) | 215 (49.3) | REF | | | |
| Overweight (25.0–29.9) | 169 (38.8) | −0.62 (−1.35 to +0.12) | | | |
| Obese (≥30.0) | 51 (11.7) | −0.25 (−1.36 to +0.86) | | | |
| | | | | | |
| Total comorbidities[f] | | | 0.704 | NI | |
| 0 | 230 (52.8) | REF | | | |
| 1–2 | 172 (39.4) | −0.22 (−0.94 to +0.49) | | | |
| 3–7 | 34 (7.8) | −0.47 (−1.81 to +0.86) | | | |
| | | | | | |
| Prior SARS-CoV-2 infection | | | <0.001 | | <0.001 |
| No | 375 (86.0) | REF | | REF | |
| Yes | 61 (14.0) | +2.26 (+1.32 to +3.20) | | +2.39 (+1.49 to +3.29) | |
| | | | | | |
| SARS-CoV-2 vaccine | | | 0.612 | | 0.386 |
| mRNA based | 300 (68.8) | −0.35 (−1.09 to +0.39) | | −0.50 (−1.20 to +0.21) | |
| Vector based | 132 (30.3) | REF | | REF | |
| Heterologous[g] | 4 (0.9) | −0.91 (−4.85 to +3.02) | | −0.50 (−4.18 to +3.18) | |
| | | | | | |
| Days between last vaccine dose and postvaccination sampling (per 10-day increase) | | −0.22 (−0.40 to −0.03) | 0.020 | −0.23 (−0.41 to −0.06) | 0.009 |
| | | | | | |
| Current CD4/CD8 ratio[f] | | | 0.618 | NI | |
| <0.50 | 21 (4.8) | +0.03 (−1.57 to +1.64) | | | |
| 0.50–0.99 | 118 (27.1) | +0.38 (−0.39 to +1.15) | | | |
| ≥1.0 | 297 (68.1) | REF | | | |
| | | | | | |
| Current CD4 count, cells/mm$^3$[f] | | | 0.157 | NI | |
| <350 | 22 (5.1) | +0.44 (−1.13 to +2.02) | | | |
| 350–499 | 47 (10.8) | +0.22 (−0.93 to +1.37) | | | |
| 500–749 | 148 (33.9) | +0.87 (+0.12 to +1.62) | | | |
| ≥750 | 219 (50.2) | REF | | | |
| | | | | | |
| Current CD8 count, cells/mm$^3$[f] | | | 0.610 | NI | |
| <350 | 109 (25.0) | REF | | | |
| 350–499 | 95 (21.8) | −0.69 (−1.70 to +0.32) | | | |
| 500–749 | 91 (20.9) | −0.24 (−1.26 to +0.77) | | | |
| ≥750 | 141 (32.3) | −0.28 (−1.19 to +0.62) | | | |

[a]General abbreviations: BMI, body mass index; CI, confidence interval; REF, reference group; NI, variable not included in multivariable analysis.
[b]Number (total $n$ = 436) and percentage of participants for each variable category.
[c]Univariable and multivariable analyses were adjusted for the prevaccination level of IFN-$\gamma$ release upon SARS-CoV-2 nucleocapsid- and spike-peptide pool stimulation.
[d]Peripheral blood mononuclear cells (PBMCs) for SARS-CoV-2 T-cell responses as measured by IFN-$\gamma$-release assay were available from 436 of 441 participants. The values represent regression coefficients ($\beta$) with 95% CI of tobit regression (censored lower bound at 0.01 pg/mL). The unit is log$_{10}$ picograms per milliliter.
[e]At the moment of the postvaccination blood draw.
[f]The last available data prior to receiving the first vaccine dose of the primary vaccination course.
[g]Received one dose of ChAdOx1 and one dose of BNT162b2.

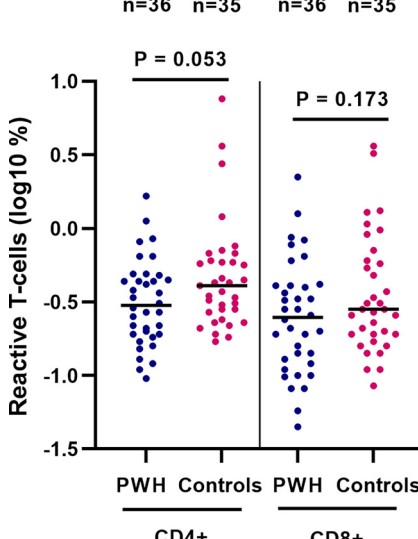

**FIG 5** SARS-CoV-2-reactive CD4+ and CD8+ T cells in 71 participants of the AGE$_h$IV COVID-19 substudy postvaccination, by HIV status. SARS-CoV-2 specific T-cell reactivity measured by activation-induced marker assay was analyzed in 71 participants. Results are expressed as the $\log_{10}$ percentage of reactive T cells. $P$ values, comparing people with HIV (PWH) and controls, were calculated using the Wilcoxon rank sum test.

associated with CD38 and HLA-DR expression on classical monocytes and positively associated with CD64 on CD16+ monocytes (Table S12). There was no significant interaction with HIV status for any of the mentioned associations. Postvaccination SARS-CoV-2 anti-S IgG titers and neutralization of ancestral SARS-CoV-2 were not associated with any of the T-cell and monocyte phenotypic markers, after adjustment for factors shown to be associated with these humoral responses in the overall study population (Tables S13 and S14).

## DISCUSSION

In this study, humoral and cellular immune responses following SARS-CoV-2 vaccination were compared between middle-aged people with longstanding but well-controlled HIV and demographically and lifestyle-comparable controls without HIV. Four to 13 weeks after completing primary vaccination, there were no significant differences between the groups in humoral immune responses, including SARS-CoV-2 antibody production by memory B cells and neutralizing antibody responses against ancestral and Delta variants of SARS-CoV-2.

Whereas these findings resemble those of a few prior studies comparing PWH on ART with a high median current CD4 cell count to controls (4–6), they differ from others reporting lower antibody responses postvaccination in PWH (7–10). Of note, in these latter studies, PWH and controls importantly differed concerning age and sex, which likely explains these seemingly discordant findings. In addition, differences in the distribution of CD4 count among PWH are also likely to be important, with some studies reporting reduced antibody responses in PWH with CD4 counts of <200 cells/mm³ (14–16).

Our results regarding neutralizing antibody responses likewise resemble those from most other studies to date (4–6, 8), although some have reported lower neutralizing titers in PWH compared to controls, again particularly in those with lower CD4 counts (7). Of note, only less than 10% of PWH included in the present study had a current CD4 count of <350 cells/mm³.

Our finding of comparable memory B-cell responses between PWH and controls seems reassuring given the importance of memory B-cell formation after vaccination for long-term protection (17).

SARS-CoV-2-specific T-cell responses as measured by IFN-γ release assay were found to be higher in PWH, both before and after vaccination, and HIV-positive status was independently associated with increased IFN-γ release following vaccination. Importantly, an in-depth flow cytometry analysis revealed no significant differences in the percentages of SARS-CoV-2-

reactive CD4$^+$ and CD8$^+$ T cells between PWH and controls, with a tendency toward a lower percentage of reactive CD4$^+$ T cells in PWH.

Other studies comparing postvaccination T-cell responses between PWH and controls have been thus far scarce and small. Most studies found no significant difference between PWH on ART and controls (5, 12, 18), but one study reported lower T-cell responses in PWH with CD4 counts of <500/mm³ but similar responses in PWH with CD4 counts of >500/mm³ compared to unmatched health care workers (14).

No detectable IFN-$\gamma$ release in response to SARS-CoV-2 peptide stimulation was observed in 56.9% of our participants after primary vaccination. It should be noted that the participants in our study are older, which has previously been associated with reduced cellular vaccine responses (19).

Our analysis of possible immune correlates of the SARS-CoV-2-specific cellular immune responses revealed that SARS-CoV-2-specific IFN-$\gamma$ release was positively associated with expression of CD163 on classical monocytes, regardless of HIV status. Given that CD163 expression on classical monocytes was significantly higher in PWH than in controls, this may explain the overall increased IFN-$\gamma$ release in PWH. The percentage of reactive CD4$^+$ T cells was negatively associated with the expression of CD163 on classical monocytes and positively associated with expression of CD163 and Fc$\gamma$ receptor 1 (CD64) on CD16$^+$ monocytes, two markers which did not differ between PWH and controls. The percentage of reactive CD8$^+$ T cells was negatively associated with the expression of CD38 and HLA-DR on classical monocytes and positively associated with the expression of CD64 on CD16$^+$ monocytes, again without significant differences in the level of these markers between PWH and controls. Expression levels of CD38, CD64, HLA-DR, and CD163 have been previously associated with monocyte activation and production of proinflammatory cytokines (20). Moreover, PBMCs obtained from PWH more often showed IFN-$\gamma$ production even in the presence of medium alone (data not shown), which resembles observations in individuals without HIV with elevated levels of immune activation possibly resulting from exposure to other infections (21). Taken together, this suggests complex relations between the activation levels of monocyte subsets, their functionality, and vaccine-induced SARS-CoV-2 T-cell responses in both PWH and controls.

Irrespective of HIV status, prior SARS-CoV-2 infection and having received a mRNA-based vaccine were the factors most strongly associated with higher postvaccination anti-S IgG titers. This resembles prior observations in both the general population (22, 23) and PWH (4, 6, 9, 15). The association between female sex and higher anti-S IgG titers is also consistent with previous reports, as well as the association between higher neutralization titers against ancestral SARS-CoV-2 and age of <60 years or receipt of an mRNA-1273 vaccine.

Prior SARS-CoV-2 infection was also associated with higher IFN-$\gamma$ release in our study, as has been reported in the general population (24). Whereas some studies have shown higher cellular responses after homologous dual-dose ChAdOx1 than BNT162b2 (25), and others showed higher responses after mRNA-based vaccines compared to ChAdOx1 (26, 27), we did not observe an association with vaccine type in our study.

Having a current CD4/CD8 ratio of <1.0, regardless of HIV status, was associated with lower anti-S and anti-RBD IgG titers, lower neutralization titers against ancestral SARS-CoV-2, and lower IFN-$\gamma$ release only in PWH. A lower CD4/CD8 ratio has been associated with the presence of a higher percentage of senescent and activated T cells, and a greater degree of immunosenescence may result in lower vaccine responses (28).

A key strength of our study is the well-characterized study population with identical assessments in both PWH and controls, who are highly comparable with regard to age, sex, and sociodemographic and behavioral characteristics. This allows for an unbiased assessment of the potential association between HIV-positive status and immune responses following SARS-CoV-2 vaccination. Furthermore, this study is one of the few to measure humoral and cellular responses both pre- and postvaccination in a cohort of this magnitude, as well as neutralization titers in a matched subgroup. Importantly, by using both questionnaire data and measuring nucleocapsid antibodies, we could determine which participants had experienced a SARS-CoV-2 infection prior to vaccination or

between pre- and postvaccination samplings, allowing us to appropriately assess the impact of hybrid compared to vaccine-induced immunity.

In terms of limitations, first our results are applicable to PWH with well-controlled viremia on ART without severe immune deficiency and may not apply similarly to PWH with a much lower CD4 count in spite of ART or those not on ART. Second, data on vaccination status and SARS-CoV-2 test results were self-reported and the precise moment of prior SARS-CoV-2 infection could not be ascertained. Hence, we could not adjust for time since SARS-CoV-2 infection. Finally, in view of the limited number of participants vaccinated with mRNA-1273 or Ad26.COV2.S, there was insufficient power to assess the responses to each of the vaccines, and we had to categorize the four vaccines into mRNA-based or vector-based vaccines.

In conclusion, our results offer reassurance to health care providers and patients in demonstrating that living with HIV, while being virally suppressed on ART with a relatively preserved CD4 count, generally does not impact short-term SARS-CoV-2 vaccine-induced or hybrid immunity to SARS-CoV-2. Determinants of vaccine responses were largely similar in both PWH and controls. Extended follow-up is needed to assess the possible impact of HIV on response durability and response to booster vaccinations. Additional larger studies, including in more heterogeneous populations of PWH with greater degrees of immune deficiency and less optimal access to antiretroviral treatment, will be required to identify correlates of protection against (severe) COVID-19.

## MATERIALS AND METHODS

**Study design and participants.** For the current study, data were collected between September 2020 (first substudy visit) until November 2021 (the last additional blood draw after completing primary vaccination) in participants of a COVID-19 substudy (13) within the AGE$_h$IV Cohort Study (29). Participants include PWH of 55 years or older and a highly comparable control group without HIV. An overview of study visits and additional information on the cohort, participant inclusion, and sample collection are shown in Fig. S2 and Text S1 in the supplemental material. Written informed consent was obtained from all participants. The study was approved by the ethics committee of the Amsterdam University Medical Centres (UMC), AMC, and is registered at www.clinicaltrials.gov (ClinicalTrials registration no. NCT01466582).

The blood sample most closely preceding the first dose of a SARS-CoV-2 vaccine (prevaccination) and the blood sample at 4 to 13 weeks after completing the primary vaccination series with BNT162b2, mRNA-1273, ChAdOx1 (two injections 4 to 12 weeks apart or a single injection for those with documented prior SARS-CoV-2 infection) or Ad26.COV2.S (single injection) were analyzed.

A subgroup of 40 PWH and 40 controls was selected for the following measurements: SARS-CoV-2 neutralization titers, percentage of reactive CD4$^+$ and CD8$^+$ T cells, and immune cell phenotyping. Inclusion criteria for this subgroup were absence of documented prior SARS-CoV-2 infection (by either self-reported positive PCR test or positive nucleocapsid [N] antibody responses by INgezim assay), vaccination with BNT162b2, mRNA-1273, or ChAdOx1 (those having received a heterologous vaccination regimen of ChAdOx1 plus BNT162b2 were excluded, as were those vaccinated with Ad26.COV2.S, due to the small sample size), and in PWH, a required current HIV-1 viral load of <50 copies/mL. Furthermore, eligible PWH were categorized in four groups based on their nadir CD4 count (using quartiles based on the CD4 nadir distribution in the total cohort of PWH). Next, 10 PWH per stratum were randomly selected and matched 1:1 to controls based on the following variables: age (using quartiles based on the distribution in the total cohort), type of vaccine, and sex at birth.

**Data collection. (i) Participant characteristics.** Data regarding date of birth, sex at birth, and ethnic origin had been obtained at time of enrollment into the parent AGE$_h$IV Cohort Study. Data on other characteristics were obtained during the last available parent study visit prior to receiving the first vaccine dose and included data on the number of prevalent comorbidities, body mass index (BMI), current CD4 and CD8 counts, last HIV test result for controls, and plasma HIV-1 RNA for PWH. PWH with HIV-1 RNA levels of <50 copies/mL or with transient viral blips of <200 copies/mL were considered virally suppressed.

In the AGE$_h$IV COVID-19 substudy, participants were asked to complete a standardized questionnaire at each substudy visit. This questionnaire included whether participants had been tested for or diagnosed with a SARS-CoV-2 infection. Information about SARS-CoV-2 vaccination status, vaccination dates, and vaccine type was also collected.

**(ii) Prior SARS-CoV-2 infection.** Participants were considered to have a prior SARS-CoV-2 infection in the case of a self-reported positive PCR test and/or a positive SARS-CoV-2 nucleocapsid (N) antibody test., allowing us to distinguish between vaccine-induced and hybrid immunity: i.e., the combined effect of prior SARS-CoV-2 infection and vaccination. The combined anti-N IgA-IgM-IgG response was measured with the semiquantitative INgezim COVID-19 double-recognition assay (Eurofins Ingenasa, Madrid, Spain) (details in Text S1). Participants who acquired SARS-CoV-2 infection between the pre- and postvaccination samplings were considered prior SARS-CoV-2 infected when assessing the postvaccination immune responses.

**Immunological endpoints (Fig. S3). (i) SARS-CoV-2 humoral immune responses.** SARS-CoV-2 anti-S and anti-RBD IgG titers were measured in serum using an in-house custom Luminex immunoassay. In short, SARS-CoV-2 S and RBD proteins were covalently coupled to Luminex MagPlex beads. Beads were incubated with 1:100,000-diluted serum overnight at 4°C. The next day, the beads were washed

followed by a 2-h incubation at room temperature with phycoerythrin (PE)-labeled goat anti-human IgG (Southern Biotech, Birmingham, AL, USA). Beads were washed, and readout was performed on a Magpix machine (Luminex, Austin, TX, USA). The resulting values are expressed as the median fluorescence intensity (MFI) of at least 50 beads per protein. Beads with tetanus toxoid and respiratory syncytial virus F protein (RSV-F) as a positive control and beads with no protein as a negative control were also included in every well. Positive- and negative-control sera were included on every plate as well as a titration of convalescent COVID-19 patient sera to monitor assay performance.

Nonresponse following SARS-CoV-2 vaccination was defined as an anti-S IgG titer of <17.8 MFI in the postvaccination sample. This was based on a cutoff value previously determined to separate PCR-confirmed COVID-19 sera and prepandemic sera with a sensitivity of 97% (95% CI, 92 to 99%) and specificity of 96% (95% CI, 94 to 98%) (30).

SARS-CoV-2 anti-S IgG production by memory B cells was determined in peripheral blood mononuclear cells (PBMCs) upon polyclonal stimulation. A total of $1 \times 10^6$ PBMCs were stimulated for 5 days with requisimod (R848; 1 $\mu$g/mL) (InvivoGen, San Diego, CA, USA) and IL-2 (10 U/mL) (Chiron Benelux, Amsterdam, The Netherlands) and cultured in RPMI 1640 culture medium supplemented with penicillin, streptomycin, and 10% heat-inactivated fetal bovine serum (FBS). Culture supernatant was harvested for determination of SARS-CoV-2 anti-S IgG production by enzyme-linked immunosorbent assay (ELISA) using SARS-CoV-2 S protein coating and anti-human IgG conjugated with horseradish peroxidase (Southern Biotech, Birmingham, AL, USA). Pooled serum of convalescent COVID-19 patients was included in each plate as a positive control. SARS-CoV-2 IgG levels in the culture medium are given relative to the positive control. The lower limit of detection was 0.5 relative units (RU) per $10^6$ PBMCs, and the lower limit of quantification was 1 RU per $10^6$ PBMCs.

SARS-CoV-2 neutralization titers against SARS-CoV-2 ancestral (USA-WA1/2020; 614D), Delta (B.1.617.2), and Omicron BA.1 (B.1.1.529) variants were measured on Vero E6 cells using an MTT [3-(4,5-dimethyl-2-thiazolyl)-2,5-diphenyl-2H-tetrazolium bromide] assay or N protein ELISA. Heat-inactivated serum was 1:2 serially diluted in $CO_2$-independent medium supplemented with L-glutamine, penicillin, streptomycin, and 10% heat-inactivated FBS, starting at a 1:20 dilution in 50 $\mu$L. An equal volume of the respective three SARS-CoV-2 strains mentioned above was added and incubated for 1 h at 37°C. Next, antibody-virus mixtures were transferred to Vero E6 cells and subsequently cultured for 48 h. Pooled serum of convalescent COVID-19 patients was included in each plate as positive control and to determine variation between plates. Neutralization of the virus was assessed using an MTT assay or SARS-CoV-2 N protein ELISA (coating antibody, 40143-R040; conjugate antibody, 40143-R001-H) (Sino Biological). Using a nonlinear regression (least-squares) curve fit, the serum dilution at which 50% of the infectivity was inhibited ($ID_{50}$) was determined (GraphPad Prism version 9.1.0; GraphPad, La Jolla, CA, USA).

**(ii) SARS-CoV-2 cellular immune responses.** SARS-CoV-2-specific T-cell responses were measured using an IFN-$\gamma$ release assay. A total of $0.5 \times 10^6$ PBMCs isolated from blood obtained at the pre- and postvaccination study visits were stimulated with a SARS-CoV-2 N- and S-peptide pool (JPT Peptide Technologies, Berlin, Germany) or cultured in medium alone as a control. After 24 h, culture supernatants were harvested and IFN-$\gamma$ released by the cells was determined by human IFN-$\gamma$ DuoSet ELISA (R&D Systems, Minneapolis, MN, USA). SARS-CoV-2-induced IFN-$\gamma$ release (picograms per milliliter) was determined by subtraction of background IFN-$\gamma$ production (medium alone) from that in the SARS-CoV-2 peptide-stimulated culture.

The activation-induced marker (AIM) assay was performed to determine the percentage of reactive T cells. PBMCs were stimulated for 6 h with a SARS-CoV-2 S-peptide pool (JPT Peptide Technologies, Berlin, Germany) at 37°C and then stained for flow cytometry. Reactive T cells were determined by coexpression of CD137 and OX40 or CD137 and CD69 within the $CD4^+$ and $CD8^+$ T-cell populations, respectively. Fluorescence was measured on the FACS Canto II fluorescence-activated cell sorter (BD Biosciences). Marker expression levels were analyzed using FlowJo version 10.8.1 (TreeStar, Ashland, OR, USA).

**(iii) Immune phenotyping of T cells and monocytes.** Immune phenotyping was performed on PBMC samples by flow cytometry (20, 31). T-cell activation was determined by coexpression of both CD38 and HLA-DR within the $CD4^+$ and $CD8^+$ T-cell populations, T-cell exhaustion was determined by expression of PD-1 within the $CD4^+$ and $CD8^+$ T-cell populations, and T-cell senescence was determined by identifying the CD27 and CD28 double-negative population within the $CD4^+$ and $CD8^+$ T-cell population. Monocyte subsets were determined by CD14 and CD16 expression: classical monocytes ($CD14^+$ $CD16^-$) and $CD16^+$ monocytes (intermediate $CD14^+$ $CD16^+$ and nonclassical $CD14^-$ $CD16^+$). Within each of these two subsets, the expression of the following monocyte activation markers was determined: CD163, CD64, CD32, CD38, and HLA-DR. To determine the expression levels of different T-cell and monocyte surface markers, the following monoclonal antibodies were used: CD163 Alexa Fluor 488 (R&D Systems, Minneapolis, MN, USA), HLA-DR fluorescein isothiocyanate (FITC), CD4 FITC, CD38 phycoerythrin (PE), CD28 peridinin chlorophyll protein (PerCP)-Cy5.5, CD4 PE-Cy7, CD64 allophycocyanin (APC)-H7, HLA-DR V500 (BD Biosiences, San Jose, CA, USA), CD32 PerCP-Cy5.5, OX40 APC, CD69 APC, CD137 APC/Fire 750, CD27 APC/Fire 750, CD8 PB, CD3 BV510 (BioLegend, San Diego, CA, USA), PD-1 PE, CD14 PE-Cy7, and CD16 eFluor450 (eBioscience). PBMCs were stained in the dark for 30 min at 4°C. Fluorescence was measured on the FACS Canto II (BD Biosciences). Marker expression levels were analyzed using FlowJo version 10.8.1 (TreeStar, Ashland, OR, USA).

**Statistical analysis.** Participant characteristics, SARS-CoV-2 anti-S and anti-RBD IgG titers, and T-cell responses were compared between PWH and controls using Pearson $\chi^2$ test, Fisher's exact test, or the Wilcoxon rank sum test, as appropriate.

The mean differences in postvaccination humoral and cellular immune responses ($\beta$) and their 95% confidence interval (95% CI) were estimated in relation to the following factors: HIV status, age, sex at birth, ethnic origin, BMI, type of vaccine (mRNA based [BNT162b2 or mRNA-1273], vector based [ChAdOx1 or Ad26.COV2.S], or heterologous [first dose, ChAdOx1; second dose, BNT162b2]), days between date of last vaccine

dose and date of sampling, evidence of prior SARS-CoV-2 infection, total number of comorbidities, current CD4 and CD8 counts, CD4/CD8 ratio, and only in PWH nadir CD4 count, years since HIV diagnosis, and years since first initiation of antiretroviral therapy (ART). These differences were modeled using linear regression for anti-S and anti-RBD IgG titers and tobit regression (censored lower bound at 0.01 pg/mL) for IFN-$\gamma$ release. Models including those with only a subgroup of matched participants also included a random intercept to account for variation between matched pairs. For all models, a multivariable model was built using a backward stepwise selection procedure, including all variables associated with a $P$ value of $<0.20$ in univariable analyses and subsequently removing all those with a $P$ value of $\geq 0.05$. HIV status and type of vaccine were forced into the model. Biologically plausible interactions between significant variables in the final multivariable model and HIV status were also assessed. Analyses were additionally stratified by HIV-status.

The association of immune cell phenotype and postvaccination IFN-$\gamma$ release, reactive CD4$^+$ T cells, reactive CD8$^+$ T cells, anti-S IgG titers, and ancestral SARS-CoV-2 neutralization was analyzed using the same regression techniques as the main analyses (i.e., linear regression, tobit regression, or linear regression with a random intercept for variation between pairs). The association was assessed unadjusted and adjusted for HIV status and for factors that in the main analyses were determined to be significantly associated with cellular or humoral responses.

Statistical significance was defined as a two-sided $P$ value of $<0.05$. Statistical analyses were carried out using Stata/IC (version 15.1) (Stata, College Station, TX, USA). Graphics were produced using GraphPad Prism (version 9.1.0; GraphPad, La Jolla, CA, USA).

**Data availability.** Data sharing has been restricted by the Ethical Review Board of the Amsterdam UMC because the data underlying this study contain very sensitive and potentially identifying information. Requests for data sharing can be made on a case-by-case basis following submission of a concept sheet as per instructions on the project website (https://agehiv.nl/en/science/). Once submitted, the proposed research or analysis will undergo review for evaluation of the scientific value, relevance to the study, design and feasibility, statistical power, and overlap of existing projects. If the proposed analysis is for verification or replication, data will then be made available. If the proposed research is for novel science, upon completion of the review, feedback will be provided to the proposers. In some circumstances, a revision of the concept may be requested. If the concept is approved for implementation, a writing group will be established consisting of the proposers (up to three people that were centrally involved in the development of the concept) and members of the AGEhIV Cohort Study Group (or other appointed cohort representatives). All persons involved in the process of reviewing these research concepts are bound by confidentiality.

## SUPPLEMENTAL MATERIAL

Supplemental material is available online only.

**SUPPLEMENTAL FILE 1**, DOCX file, 0.9 MB.

## ACKNOWLEDGMENTS

This work was supported in part through an investigator-initiated study grant from ViiV Healthcare. The parent AGE$_h$IV Cohort Study was supported by The Netherlands Organization for Health Research and Development (ZonMW, [grant no. 30002000] and AIDS Fonds [grant no. 2009063]) and in part by unrestricted research grants from Gilead Sciences, ViiV Healthcare; Janssen Pharmaceuticals N.V., and Merck Sharp & Dohme Corp. None of these funding bodies had a role in the design or conduct of the study, the analysis and interpretation of the results, the writing of the report, or the decision to publish.

All authors contributed *Conceptualization* and/or *Design*; M.L.V., L.v.P., M.G., and A.B. contributed *Methodology* and *Formal Analysis*; M.L.V., L.v.P., M.G., A.C.v.N., K.A.v.D., K.T., J.v.R., and L.v.d.H. contributed *Investigation*; M.L.V., L.v.P., and M.G. contributed *Data Curation*, *Validation*, *Visualization*, and *Writing – Original Draft*; P.R. contributed *Funding Acquisition*; P.R., N.A.K., and M.J.v.G. contributed *Supervision*. All authors contributed *Writing – Review and Editing*.

F.W.N.M.W. has served on scientific advisory boards for ViiV Healthcare and Gilead sciences. M.F.S.v.d.L. has received independent scientific grant support from Sanofi Pasteur, MSD Janssen Infectious Diseases and Vaccines, and Merck & Co., has served on advisory boards of GlaxoSmithKline and Merck & Co., and has received nonfinancial support from Stichting Pathologie Onderzoek en Ontwikkeling. M.v.d.V. through his institution has received independent scientific grant support and consultancy fees from AbbVie, Gilead Sciences, MSD, and ViiV Healthcare, for which honoraria were all paid to his institution. P.R. through his institution has received independent scientific grant support from Gilead Sciences, Janssen Pharmaceuticals, Inc., Merck & Co., and ViiV Healthcare and has served on scientific advisory boards for Gilead Sciences, ViiV

Healthcare, and Merck & Co., honoraria for which were all paid to his institution. M.L.V., L.v.P., M.G., A.B., A.C.v.N., K.A.v.D., K.T., J.v.R., M.B., L.v.d.H., M.J.v.G., and N.A.K. declare no competing interests.

The following are members of the AGEhIV Cohort Study Group. The following contributed scientific oversight and coordination: P. Reiss (principal investigator) (Amsterdam University Medical Centers, University of Amsterdam [Amsterdam UMC], Department of Global Health and Amsterdam Institute for Gobal Health and Development [AIGHD]), F. W. N. M. Wit (Amsterdam UMC, Department of Global Health and AIGHD), M. van der Valk (Amsterdam UMC, Department of Global Health and AIGHD), A. Boyd (Amsterdam UMC, Department of Global Health and AIGHD), M. L. Verburgh (Amsterdam UMC, Department of Global Health and AIGHD), I. A. J. van der Wulp (Amsterdam UMC, Department of Global Health and AIGHD), M. C. Vanbellinghen (Amsterdam UMC, Department of Global Health and AIGHD), C. J. van Eeden (Amsterdam UMC, Department of Global Health and AIGHD), M. F. Schim van der Loeff (co-principal investigator) (Public Health Service of Amsterdam, Department of Infectious Diseases), J. C. D. Koole (Public Health Service of Amsterdam, Department of Infectious Diseases), L. del Grande (Public Health Service of Amsterdam, Department of Infectious Diseases), and I. Agard (Public Health Service of Amsterdam, Department of Infectious Diseases). The following contributed to data management: S. Zaheri, M. M. J. Hillebregt, Y. M. C. Ruijs, D. P. Benschop, and A. el Berkaoui (HIV Monitoring Foundation). The following contributed to statistical support: A. Boyd (HIV Monitoring Foundation) and F. W. N. M. Wit (HIV Monitoring Foundation). The following contributed central laboratory support: N. A. Kootstra (Amsterdam UMC, Laboratory for Viral Immune Pathogenesis and Department of Experimental Immunology), A. M. Harskamp-Holwerda (Amsterdam UMC, Laboratory for Viral Immune Pathogenesis and Department of Experimental Immunology), I. Maurer (Amsterdam UMC, Laboratory for Viral Immune Pathogenesis and Department of Experimental Immunology), M. M. Mangas Ruiz (Amsterdam UMC, Laboratory for Viral Immune Pathogenesis and Department of Experimental Immunology), B. D. N. Boeser-Nunnink (Amsterdam UMC, Laboratory for Viral Immune Pathogenesis and Department of Experimental Immunology), and O. S. Starozhitskaya (Amsterdam UMC, Laboratory for Viral Immune Pathogenesis and Department of Experimental Immunology), L. van der Hoek (Amsterdam UMC, Department of Medical Microbiology and Infection Prevention, Laboratory of Experimental Virology), M. Bakker (Amsterdam UMC, Department of Medical Microbiology and Infection Prevention, Laboratory of Experimental Virology), and M. J. van Gils (Amsterdam UMC, Department of Medical Microbiology and Infection Prevention, Laboratory of Experimental Virology). The following contributed project managment and administrative support: L. Dol (AIGHD) and G. Rongen (AIGHD). The following were participating HIV physicians and nurses: S. E. Geerlings (Amsterdam UMC, Division of Infectious Diseases), A. Goorhuis (Amsterdam UMC, Division of Infectious Diseases), J. W. R. Hovius (Amsterdam UMC, Division of Infectious Diseases), F. J. B. Nellen (Amsterdam UMC, Division of Infectious Diseases), J. M. Prins (Amsterdam UMC, Division of Infectious Diseases), T. van der Poll (Amsterdam UMC, Division of Infectious Diseases), M. van der Valk (Amsterdam UMC, Division of Infectious Diseases), W. J. Wiersinga (Amsterdam UMC, Division of Infectious Diseases), M. van Vugt (Amsterdam UMC, Division of Infectious Diseases), G. de Bree(Amsterdam UMC, Division of Infectious Diseases), B. A. Lemkes (Amsterdam UMC, Division of Infectious Diseases), V. Spoorenberg (Amsterdam UMC, Division of Infectious Diseases), F. W. N. M. Wit (Amsterdam UMC, Division of Infectious Diseases), J. van Eden (Amsterdam UMC, Division of Infectious Diseases), F. J. J. Pijnappel (Amsterdam UMC, Division of Infectious Diseases), A. Weijsenfeld (Amsterdam UMC, Division of Infectious Diseases), S. Smalhout (Amsterdam UMC, Division of Infectious Diseases), I. J. Hylkema-van den Bout (Amsterdam UMC, Division of Infectious Diseases), C. Bruins (Amsterdam UMC, Division of Infectious Diseases), and M. E. Spelbrink. The other collaborators were P. G. Postema (Amsterdam UMC, Department of Cardiology), P. H. L. T. Bisschop (Amsterdam UMC, Division of Endocrinology and Metabolism), E. Dekker (Amsterdam UMC, Department of Gastroenterology), N. van der Velde (Amsterdam UMC, Division of Geriatric Medicine), R. Franssen (Amsterdam UMC, Division of Geriatric

Medicine), J. M. R. Willemsen (Amsterdam UMC, Division of Nephrology), L. Vogt (Amsterdam UMC, Division of Nephrology), P. Portegies (Amsterdam UMC, Department of Neurology), G. J. Geurtsen (Amsterdam UMC, Department of Neurology), I. Visser (Amsterdam UMC, Department of Psychiatry), A. Schadé (Amsterdam UMC, Department of Psychiatry), P. T. Nieuwkerk (Amsterdam UMC, Department of Medical Psychology), R. P. van Steenwijk (Amsterdam UMC, Department of Pulmonary Medicine), R. E. Jonkers (Amsterdam UMC, Department of Pulmonary Medicine), C. B. L. M. Majoie (Amsterdam UMC, Department of Radiology), M. W. A. Caan (Amsterdam UMC, Department of Radiology), B. J. H. van den Born (Amsterdam UMC, Division of Vascular Medicine), E. S. G. Stroes (Amsterdam UMC, Division of Vascular Medicine), and S. van Oorspronk (HIV Vereniging, Nederland).

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
