## [Reviewer comments · Microbiology Spectrum]

Microbiology Spectrum

Robust vaccine-induced as well as hybrid B- and T-cell immunity across SARS-CoV-2 vaccine platforms in people with HIV

Myrthe Verburgh, Lisa van Pul, Marloes Grobben, Anders Boyd, Ferdinand Wit, Ad van Nuenen, Karel van Dort, Khadija Tejjani, Jacqueline van Rijswijk, Margreet Bakker, Lia van der Hoek, Maarten Schim van der Loeff, Marc van der Valk, Marit van Gils, Neeltje Kootstra, and Peter Reiss

Corresponding Author(s): Myrthe Verburgh, Amsterdam UMC Locatie AMC

Review Timeline:

Submission Date:

March 20, 2023

Accepted:

April 22, 2023

Editor: Oliver Laeyendecker

Reviewer(s): Disclosure of reviewer identity is with reference to reviewer comments included in decision letter(s). The following individuals involved in review of your submission have agreed to reveal their identity: Yanmin Wan (Reviewer #1)

Transaction Report:

DOI: <https://doi.org/10.1128/spectrum.01155-23>

April 22, 2023

Dr. Myrthe Lauriëtte Verburgh
Amsterdam UMC Locatie AMC
Infectious Diseases
Meibergdreef 9
Amsterdam
Netherlands

Re: Spectrum01155-23 (Robust vaccine-induced as well as hybrid B- and T-cell immunity across SARS-CoV-2 vaccine platforms in people with HIV)

Dear Dr. Myrthe Lauriëtte Verburgh:

Your manuscript has been accepted, and I am forwarding it to the ASM Journals Department for publication. You will be notified when your proofs are ready to be viewed.

Sincerely,

Oliver Laeyendecker
Editor, Microbiology Spectrum
